# The Linearity of the Effect of Surprisal on Reading Times across Languages

**Weijie Xu**
University of California, Irvine
weijie.xu@uci.edu

**Jason S. Chon**
Stanford University
jchon5@stanford.edu

**Tianran Liu**
University of Washington
tianranl@uw.edu

**Richard Futrell**
University of California, Irvine
rfutrell@uci.edu

## Abstract

In psycholinguistics, surprisal theory posits that the amount of online processing effort expended by a human comprehender per word positively correlates with the surprisal of that word given its preceding context. In addition to this overall correlation, more importantly, the specific quantitative form taken by the processing effort as a function of surprisal offers insights into the underlying cognitive mechanisms of language processing. Focusing on English, previous studies have looked into the linearity of surprisal on reading times. Here, we extend the investigation by examining eye-tracking corpora of seven languages: Danish, Dutch, English, German, Japanese, Mandarin, and Russian. We find evidence for superlinearity in some languages, but the results are highly sensitive to which language model is used to estimate surprisal.

## 1 Introduction

A great deal of insight into human language processing can be gleaned by studying the word-by-word processing difficulty experienced by a comprehender who is listening to or reading language. Consequently, the timecourse of incremental language processing difficulty has formed one of the major objects of study in psycholinguistics (Sedivy, 2014, Ch. 8). Here we take up the question of the functional form of the effect of **surprisal** (the negative log probability of each word given preceding context) on word-by-word processing difficulty, as indexed by reading time. Whereas previous empirical work has addressed this question in English, we study the question in the existing reading time databases of Danish, Dutch, English, German, Japanese, Mandarin Chinese, and Russian. To preview the results, we find consistent evidence for superlinearity for German and Russian; for all other languages except Mandarin, we find a linear effect of surprisal, but the analysis is inconclusive about the presence of additional superlinear effects

and the result is highly sensitive to which language model is used to estimate surprisal.

## 2 Background

In the field of psycholinguistics, **Surprisal Theory** (Hale, 2001; Levy, 2008) holds that processing difficulty is information-theoretic in nature. Letting $w_t$ be the word that a comprehender is perceiving at time $t$, and $w_{<t}$ be the sequence of all previous words, the surprisal $S_t$ measures the amount of information contained in $w_t$ in context:

$$S_t \equiv -\log p\left(w_t \mid w_{<t}\right). \quad (1)$$

In general, Surprisal Theory holds that the amount of processing difficulty (for our purposes, the reading time, or RT) experienced by a comprehender for word $w_t$ in context $w_{<t}$ is given by $f(S_t)$, where $f$ is a monotonically increasing **linking function** converting bits of information into milliseconds of RT.

Surprisal Theory is well-supported empirically by findings of a correlation between surprisal and RT, and the ability of surprisal to explain RT effects previously attributed to other processing mechanisms (Levy, 2008), although it may not explain the full range of human processing difficulty (van Schijndel and Linzen, 2021; Arehalli et al., 2022). The correlation between surprisal and RT gets stronger when probabilities are calculated using lower-perplexity language models (Goodkind and Bicknell, 2018; Hao et al., 2020; Wilcox et al., 2020), although this relationship may not be universal across languages (Kuribayashi et al., 2021), and breaks down for recent large models (Oh et al., 2022; Oh and Schuler, 2023), whose ability to predict words is super-human (Buck et al., 2022).

The goal of this work is to empirically investigate the form of the linking function $f$ across languages. Although early work did not commit to a specific form for the linking function $f$ (Hale,

| Language | Corpus | Genre | Tokens | Subjects |
|----------|--------|-------|--------|----------|
| Danish | Copenhagen Corpus (Hollenstein et al., 2022) | Speeches | 26,454 | 57 |
| Dutch | Ghent Eyetracking Corpus (GECO; Cop et al., 2017) | Novel | 58,302 | 18 |
| English | Dundee Corpus (Kennedy et al., 2003) | Editorials | 24,679 | 10 |
| German | Potsdam Sentence Corpus (Kliegl et al., 2006) | Artificial | 557 | 273 |
| Japanese | BCCWJ-EyeTrack (Asahara et al., 2016) | News | 970 | 24 |
| Mandarin | Beijing Sentence Corpus (Yan et al., 2010) | Artificial | 1,521 | 30 |
| Russian | Russian Sentence Corpus (Laurinavichyute et al., 2019) | Artificial | 892 | 96 |

Table 1: Eyetracking corpora used as datasets. 'Artificial' genre refers to sentences written by experimenters and presented in isolation. 'Tokens' indicates the number of text tokens (as defined using the tokenization provided by the eyetracking corpus) for which there is reading time data from at least one subject after all data exclusions.

2001; Levy, 2005), subsequent theoretical and empirical developments have suggested that the linking function $f$ might be a simple proportionality, that is $f(S_t) = k \cdot S_t$, where $k$ is a scalar conversion factor from bits of information to milliseconds of RT (Levy, 2008; Smith and Levy, 2013). Several studies have found evidence for linearity of the linking function $f$ using both $n$-gram models and modern language models (Smith and Levy, 2013; Goodkind and Bicknell, 2018; Merkx and Frank, 2021; Aurnhammer and Frank, 2019; Wilcox et al., 2020; Shain et al., 2022).

However, the picture is complicated by theoretical considerations and recent empirical evidence. Theoretically, a superlinear function is favored by a model of incremental language processing in which an incremental parse tree is derived from a sentence prefix by sampling (Hoover et al., 2023), and the possibility of a superlinear linking function is related to the observation that speakers tend to communicate information at a relatively constant rate per unit time (Fenk and Fenk, 1980; Levy and Jaeger, 2007; Jaeger, 2010; Clark et al., 2023): such effects can be predicted from Surprisal Theory if processing cost is superlinear in surprisal (Smith and Levy, 2013; Levy, 2005, 2018). Empirically, Meister et al. (2021) find evidence for a small superlinear effect of surprisal on reading times at the level of sentences, and Hoover et al. (2023) find superlinear effects at the level of words.

Our work addresses the fact that broad-coverage psycholinguistic studies of surprisal have for the most part used reading time corpora of English only, and the functional form of the effect of surprisal on word RT in particular has only been investigated outside of English in one other study (Wilcox et al., 2023, to be discussed below in Section 5) to our knowledge.

## 3 Methods

### 3.1 Reading-time data

We use the reading-time datasets described in Table 1. There are two classes of reading-time corpus. The first is corpora of large amounts of naturalistic connected text, such as the English Dundee corpus, the Dutch GECO corpus, the Danish CopCo corpus, and the Japanese BCCWJ-EyeTrack; these corpora contain many different tokens in different contexts. The second class is the 'Sentence Corpora' of Mandarin, German, and Russian; these corpora contain only a few hundred artificially-constructed sentences which are read in isolation, but they have data from many participants.

Following Smith and Levy (2013), we examine first-pass gaze durations per word when this is possible based on the available eyetracking data; otherwise we examine first fixation duration. First-pass gaze duration, also called first-pass reading time and first-run dwell time, is defined as the sum of the duration of all fixations on a word $w$, starting with the first fixation on $w$ and ending with any fixation on any other word.[1]

**Data exclusions.** Following Smith and Levy (2013), we exclude the following datapoints from our analysis: words at the beginning or end of a line (when this could be determined in a corpus); words attached to punctuation[2]; and words that were skipped or have an RT of 0 ms. We also ex-

---

[1] For the English Dundee corpus, we used the sum of FDUR values in the first pass. For Dutch, we used the WORD_FIRST_FIXATION_TIME field. For German, we used the dur field. For Mandarin, we used the fixation_duration field. For Danish, we used the word_first_pass_dur field. For Japanese, we used the fpt field. For Russian, we used the IA_FIRST_DWELL_TIME field.

[2] This exclusion does not affect the Sentence Corpora, where words were presented without punctuation.

| Language | Train Size | Dev. PPL |
|----------|-----------|----------|
| Danish | 42M | 46.3 |
| Dutch | 166M | 32.9 |
| English | 1,856M | 32.6 |
| German | 831M | 31.8 |
| Japanese | 292M | 52.2 |
| Mandarin | *804M* | *3.1* |
| Russian | 451M | 32.6 |

Table 2: Summary of the training set size and the dev set perplexity of the monolingual surprisal models trained in Wikipedia text. Training set sizes are per-token according to our trained tokenizers. The training set size and dev set perplexity for Mandarin are italicized to indicate that these are per-byte, rather than per-word, reflecting the byte-level tokenization of our Mandarin language model.

clude datapoints with a surprisal value greater than 20 nats or an RT greater than 2000 ms.

### 3.2 Language models

We use two sets of language models to derive surprisal estimates. First, we use the existing mGPT model (Shliazhko et al., 2022), a large multilingual model trained on Wikipedia text also used by Wilcox et al. (2023). Second, we train new autoregressive language models using the GPT-2 architecture (Radford et al., 2019) on text from Wiki40b dataset (Guo et al., 2020), removing title and section headers and splitting the remaining content by paragraph. We use the provided training/dev set splits. For all languages except Japanese and Mandarin, we train the default GPT-2 tokenizer on the training set. For Japanese, the GPT-2 tokenizer uses excessive memory, so we use the XLNet tokenizer instead (Yang et al., 2019). For Mandarin, we tokenize the text by byte (Xue et al., 2022), because the GPT-2 tokenizer splits up the sentences in the eyetracking corpus very differently from the provided tokenization.

We train language models to minimize cross-entropy loss on the training set using the `huggingface` library (Wolf et al., 2020). For languages other than Mandarin, we continue training until either dev set loss reaches 3.5 nats, or 7 epochs through the training data have been completed. For Mandarin, we continue training until the decrease in dev set loss is flat by visual inspection; the special treatment of Mandarin is because its byte-level loss is not comparable to the token-level loss for

the other languages. Table 2 shows training set size and final language model perplexities.[3]

### 3.3 Statistical Analysis

We analyze the RT data by fitting linear mixed-effects models (Baayen et al., 2008) to predict RT as a function of surprisal, controlling for word length. In order to test for a linear effect of RT, we fit a sequence of models:

1. $M_0$: A regression including word length and log-transformed word frequency from Speer (2022) as a control predictor, with random intercepts by subject. This is the maximal consistently converging random effect structure (Barr et al., 2013).

2. $M_L$: $M_0$ plus a linear effect of surprisal.

3. $M_{LQ}$: $M_L$ plus a quadratic effect of surprisal, testing for a superlinear effect beyond the linear effect.

The existence of a linear effect was determined by a likelihood ratio test comparing $M_0$ to $M_L$. The existence of an additional superlinear effect was tested by comparing $M_L$ to $M_{LQ}$.

**Spillover.** In reading time data, processing slowdown due to factors such as surprisal is often delayed, appearing on following words (Erlich and Rayner, 1983). In order to control for this effect, we adopt the standard practice in psycholinguistics: for all the variables in our models, we use not only the value of the current word, but also the value of the $K$ previous words, where the value of the spillover window size $K$ is determined by the following procedure: first a control model is fit to the reading time data using only word length as a predictor, and then a second model is fit using the same variables, incrementing the spillover window size by one. If the incremented model is a significantly better fit to the data than the control model by a likelihood ratio test, then the size of the spillover window is expanded. The resulting spillover window sizes for the four corpora are shown in Table 3. If a token does not have valid surprisal values for all previous tokens within its spillover window, it is excluded from statistical analysis. Results following an alternative procedure where the spillover window is fixed at 2 for all corpora are shown in Appendix A.

---

[3]The trained LMs are available at `https://huggingface.co/rfutrell`.

| Language | Spillover | mGPT | | | Monolingual | | |
|---|---|---|---|---|---|---|---|
| | | ET PPL | Linear | Quadratic | ET PPL | Linear | Quadratic |
| Danish | 3 | 41.1 | + | $p = 0.427$ | 76.7 | + | + |
| Dutch | 3 | 124.5 | + | − | 295.7 | + | + |
| English | 2 | 52.0 | + | $p = 0.935$ | 74.9 | + | + |
| German | 3 | 37.1 | + | + | 186.9 | + | + |
| Japanese | 0 | 16.1 | + | $p = 0.354$ | 215.9 | + | $p = 0.326$ |
| Mandarin | 0 | 21.4 | $p = 0.275$ | $p = 0.487$ | *30.7* | $p = 0.284$ | $p = 0.081$ |
| Russian | 3 | 27.1 | + | + | 220.6 | + | + |

Table 3: Summary of results using the mGPT language model and the monolingual models trained in Wikipedia text to estimate surprisal. 'Spillover' is the spillover window size determined by the procedure in Section 3.3. 'ET PPL' is the perplexity of each language model evaluated on the eyetracking corpus text. Note that the perplexities across models are not directly comparable because of different tokenizations. The monolingual perplexity for Mandarin is italicized to indicate that these are per-byte, rather than per-word, reflecting the byte-level tokenization of our Mandarin language model. 'Linear' is the sign of any significant (at $p < 0.05$) linear effect of surprisal on RT for the current token, and 'Quadratic' is the sign of any significant quadratic effect for the current token; $p$ values are presented if the effect is not significant.

## 4   Results

Figure 1 shows average reading time as a function of surprisal, along with linear fits, using the mGPT surprisal estimates. A strong positive relationship always obtains. Visually, the effect appears to be largely linear in the larger corpora of Danish, Dutch, and English, but the shape of the effect is harder to discern for the the smaller corpora (German, Japanese, Mandarin, and Russian).

Regression results for both surprisal models are schematized in Table 3, which gives the sign of the coefficient for (linear or quadratic) surprisal in the fitted regression predicting reading time. As expected, the sign of the linear effect is always positive, meaning that increasing surprisal corresponds to increasing reading time; the effect is significant in all languages except Mandarin. For the mGPT model, the quadratic effect is significantly negative in Dutch, indicating sub-linearity, and significantly positive in German and Russian, indicating superlinearity. The quadratic effect is not significant elsewhere. For the monolingual models, we find positive quadratic effects in all languages except Japanese and Mandarin.

The full regression code and results are available at `https://github.com/weijiexu-charlie/Linearity-of-surprisal-on-RT`.

### 4.1   Discussion

Overall, the results provide evidence for a positive linear effect of surprisal on reading time across languages, and inconsistent evidence for a superlinear effect: the quadratic effect of surprisal varies in significance and sign depending on the language model used.

It is hard to discern a systematic difference between languages showing evidence for only a linear effect and those showing possible deviations from linearity—a consistent superlinear effect is found only in German and Russian, the two smallest datasets. The evidence for superlinearity does not seem to follow a typological pattern.

## 5   Concurrent work

Concurrent to our work, Wilcox et al. (2023) have examined the linearity of the effect of surprisal on RT in 11 languages of the MECO eyetracking corpus (Siegelman et al., 2022), finding consistent evidence for linearity and a lack of evidence for superlinearity across all languages studied.[4] Our work is complementary with this other work, differing in datasets, languages considered, and statistical methodology. Regarding data, Wilcox et al. (2023) use the MECO corpus, consisting of parallel translated text, which is more similar in size to the 'Sentence Corpora' that we use for German, Mandarin, and Russian. In contrast, we use larger corpora of naturalistic text for Danish, Dutch, English, and Japanese. We also investigate reading times in Japanese and Mandarin, which

---

[4] Another related work is de Varda and Marelli (2022, 2023), who study the existence of a surprisal effect across languages; however, they do not test the functional form of surprisal.

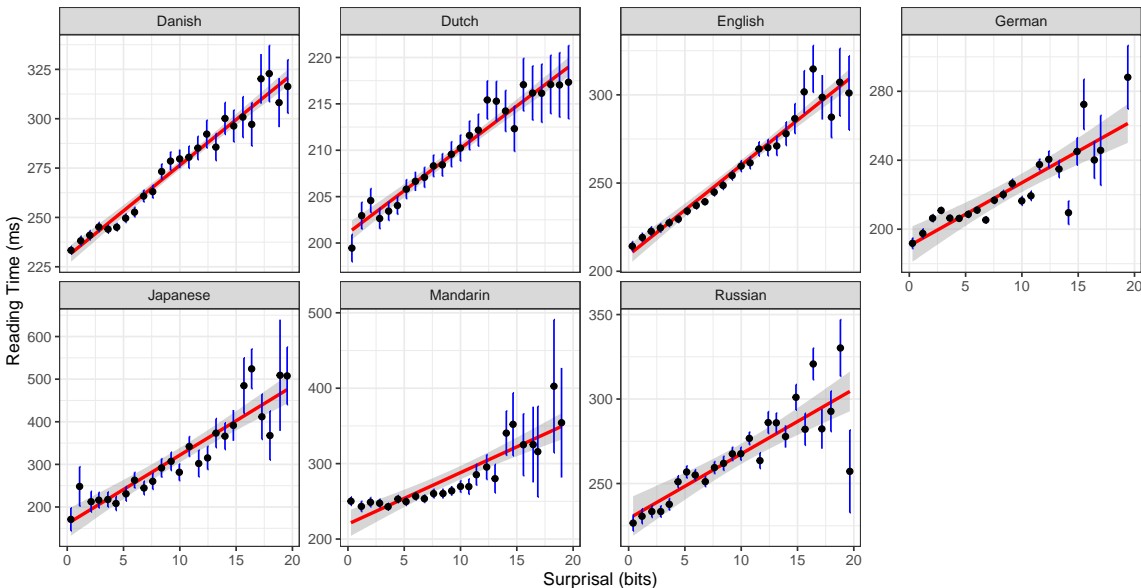

Figure 1: Average RT as a function of the surprisal from mGPT. Surprisal is binned into 25 categories, and the mean RT within each category is shown in black with a 95% confidence interval in blue. A linear fit to these points is shown in red.

are not present in Wilcox et al. (2023). Reading times in these languages are especially interesting because they are written using orthographies that do not mark word boundaries; thus reading in these languages involves some arguably different cognitive processes than the other languages, as word/morpheme segmentation must be inferred from the text.

Our work also differs from Wilcox et al. (2023) in the statistical methodology. Following Smith and Levy (2013) and Wilcox et al. (2020), Wilcox et al. (2023) compare fits of linear and non-linear Generalized Additive Models when predicting reading times from surprisal after controlling for word length and frequency, with a fixed spillover window across languages. In contrast, we adopt a simpler approach: we compare a mixed-effects linear model against a quadratic model, with a variable spillover window across languages.

Resolving the apparent tension between our results, which join Hoover et al. (2023) in finding possible evidence for superlinearity, against the results suggesting pure linearity (Shain et al., 2022; Wilcox et al., 2023, for example) should be a focus in future research.

## 6 Conclusion

We conclude that the effect of surprisal on reading times across languages may be slightly superlinear, but the evidence for this superlinearity is unstable

and depends on the language model used. Determining the functional form of the effect in further languages will require reading time corpora of large amounts of naturalistic text and strong language models.

## Limitations

The generality of our conclusions is limited by factors related to statistical methodology, the nature of the corpora used, and the nature of the language models trained.

**Statistical Methodology**   Our analysis is limited because we adopt a relatively simple statistical methodology, in three ways.

First, our handling of spillover effects, while standard in the psycholinguistic literature, does not account for the full shape of the timecourse of the effect of surprisal, which may lead to a lack of power to detect effects (Shain and Schuler, 2018). More sophisticated methods for handling spillover effects, such as deconvolutional time series methods (Shain, 2019; Shain et al., 2022), may yield different results. The procedure of setting a fixed spillover window also limits statistical power because a token can only be kept in the regression if it has valid surprisal values for all previous tokens within the spillover window; the result is that many tokens must be thrown out.

Second, our test for superlinear effects of sur-

prisal on RT is relatively simple: we compare a model with only a linear effect against a model with a linear+quadratic effect. In contrast, Smith and Levy (2013) search for linearity using a method involving cubic splines, and Meister et al. (2021) fit models predicting RT as

$$\text{RT} = k \cdot S_t^{\alpha} \qquad (2)$$

for varying $\alpha$, allowing them to quantify the shape of the superlinearity (they find best overall fits with values between $\alpha = 1$ and $\alpha = 1.5$). Our method can detect the existence of a superlinear effect but cannot find a specific exponent $\alpha$ to characterize it.

Third, we use linear regression to predict reading times, because we want to study a linear effect of surprisal on RT in particular. However, linear regression assumes a Gaussian error distribution, which does not generally hold for human reaction times: human reaction times cannot be negative and tend to be skewed to the right, and have been characterized using distributions such as Log-Normal (Baayen and Milin, 2010) and Ex-Gaussian (Luce, 1986). The use of a more realistic error distribution may lead to different results, especially in our critical datapoints where RT values are higher than expected from linear Surprisal Theory.

**Eyetracking Corpora** We use publically-available eyetracking datasets which are not matched across languages, because the datasets were developed by different researchers for different purposes at different times. As a result, the corpora differ not only in language but also genre.

**Language Models** We only use one language model architecture (GPT), which is a potential source of inaccuracy in our surprisal values: other language model architectures may deliver surprisal values that better reflect human expectations. Furthermore, the ET corpus texts differ in genre from the Wikipedia text corpora used to train the language models. This discrepancy may result in surprisal values for the ET corpus texts that do not reflect human expectations. This factor may account for the discrepancy where we find a superlinear effect in English whereas some previous work has not (for example, Wilcox et al., 2020).

Another potential limitation is that we train all the monolingual language models to the same target perplexity, which may not reflect the same level of language model quality per language because different languages may have different inherent entropy rates (Bentz et al., 2017).

## Ethics Statement

We do not foresee ethical issues with this work.

## Acknowledgements

We thank Masayuke Asahara, Nora Hollenstein, Reinhold Kliegl, Anna Laurinavichyute, and Ming Yan for generously sharing data and answering our questions. This work was supported by NSF Grant #1947307 and an NVIDIA GPU Grant and to R.F.

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

## A   Results with a Fixed Spillover Window

Table 4 shows our main results but using a fixed spillover window of size 2 for all languages, for the mGPT and monolingual surprisal models. When using mGPT, fixing the spillover window yields a significant positive linear effect of surprisal in Mandarin, eliminates the evidence for the apparent sublinear effect in Dutch, and gives evidence for a superlinear effect in Japanese. When using the monolingual models, the results are the same as in the main text, but now with an apparent positive quadratic effect in Japanese.

| Language | Linear | | Quadratic | |
|---|---|---|---|---|
| | mGPT | Monoling | mGPT | Monoling |
| Danish | + | + | ns, $p = 0.289$ | + |
| Dutch | + | + | ns, $p = 0.101$ | + |
| English | + | + | ns, $p = 0.935$ | + |
| German | + | + | + | + |
| Japanese | + | + | + | + |
| Mandarin | + | ns, $p = 0.058$ | ns, $p = 0.855$ | ns, $p = 0.424$ |
| Russian | + | + | + | + |

Table 4: Main results as in Table 3 using the mGPT surprisal model with afixed spillover window of size 2.

## B  Additional Figures

Figure 2 shows reading times as a function of surprisal based on the monolingual surprisal models. Figures 3 and 4 show residual reading times after controlling for spilled-over length and frequency, as a function of surprisal, for the mGPT and monolingual language models respectively.

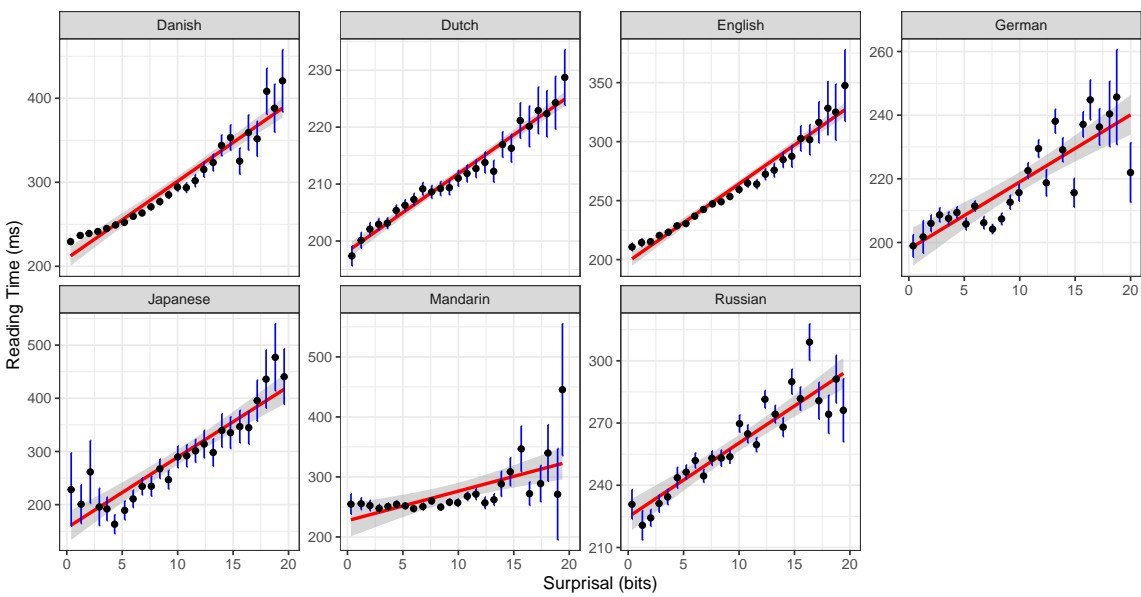

Figure 2: As in Figure 1, average RT as a function of the surprisal from monolingual LM.

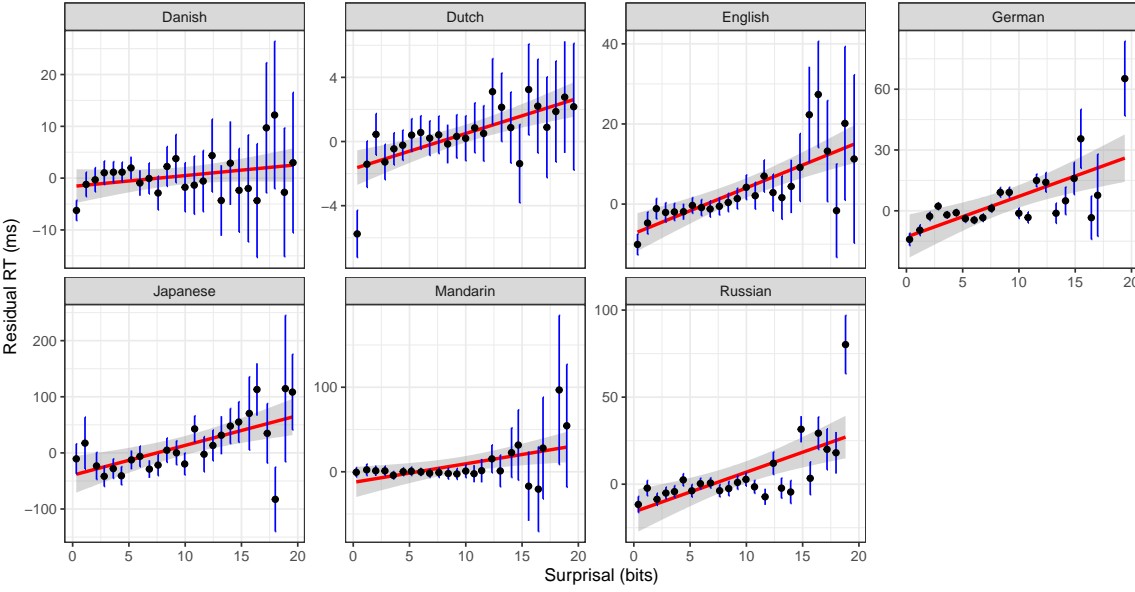

Figure 3: Average residual RT controlled by word length and word frequency as a function of the surprisal from mGPT.

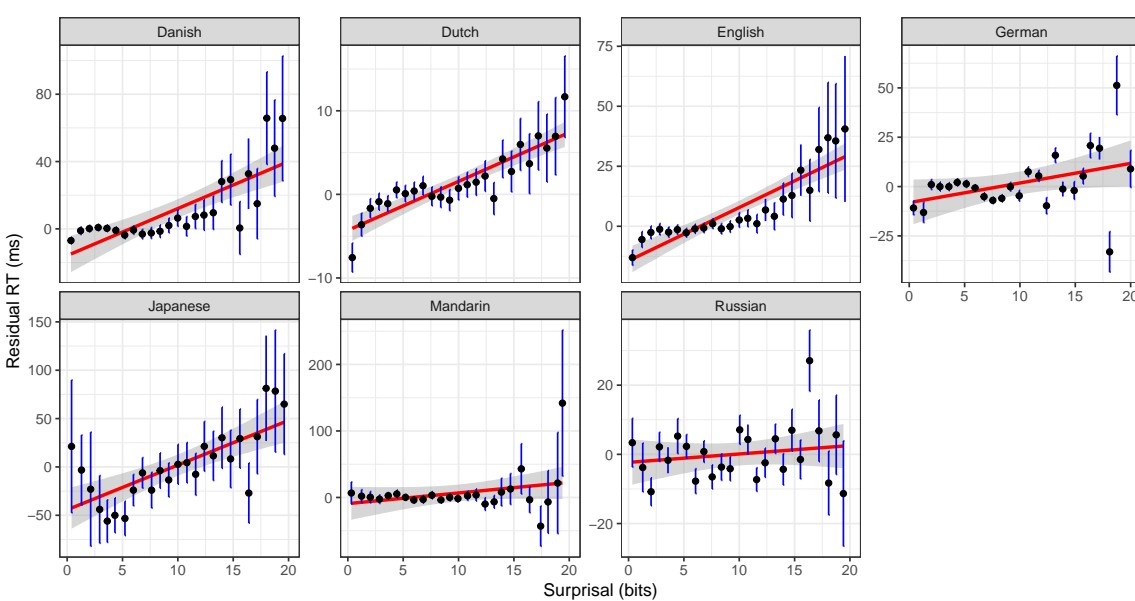

Figure 4: Average residual RT controlled by word length and word frequency as a function of the surprisal from monolingual LM.