# OpenReview forum: "The Linearity of the Effect of Surprisal on Reading Times across Languages"
_EMNLP/2023/Conference — EMNLP 2023 Findings_

### Official Review · Reviewer_fm4s · 2023-08-03

**Soundness:** 3

**Excitement:**

4: Strong: This paper deepens the understanding of some phenomenon or lowers the barriers to an existing research direction.

**Missing References:**

(a) I’m sure the authors are aware of Wilcox et al., “Testing the Predictions of Surprisal Theory in 11 Languages” but just in case you are not, then this paper was arXived recently. It reaches many of the same conclusions, but some different ones. Discussion of the differences – both in terms of methods and conclusions – will be essential in the final version of this paper.

(b) De Varda, Andrea, and Marco Marelli. "Scaling in Cognitive Modelling: a Multilingual Approach to Human Reading Times." Proceedings of the 61st Annual Meeting of the Association for Computational Linguistics (Volume 2: Short Papers). 2023.

(c) Andrea de Varda and Marco Marelli. 2022. The effects of surprisal across languages: Results from native and non-native reading. In Findings of the Association for Computational Linguistics: AACL-IJCNLP 2022, pages 138–144, Online only. Association for Computational Linguistics.


**Paper Topic And Main Contributions:**

Summary: The authors investigate the functional form of the surprisal / reading time relationship across languages, honing in on whether the link is linear or super linear, which has been a topic of some debate in recent literature. Previous work has addressed this question either through visual inspection of the surprisal / RT function or through linear vs. nonlinear model comparison. The authors employ the latter, albeit a slightly different method than has been used in previous studies. They find that about half the languages tested demonstrate superlinear surprisal / RT effects, and that these tend to be for languages with larger datasets. Overall, I think this paper contributes to the current discussion and that the community will benefit from its inclusion in EMNLP. However, I do have some concerns about the statistical tests used.

**Questions For The Authors:**

(a) (135) It would be very helpful to see which languages are first fixation and which are gaze durations. It’s a little difficult to tell from Footnote 1. Relatedly, was there a relationship between FF / GD in the linear / non-linearity of the results? How do the authors think the different metrics changed or impacted their results? I would appreciate more discussion about this.

(b) (147) Could the authors provide a justification for this? Is this something that has been done in the previous literature?

(c) (Fig 1) One concern I have about the figure is that plotting the raw surprisal vs. RT data does not take into account things like the effect of word length on reading times. In previous studies such as Smith and Levy (2013) and Hoover et al., (2022) figures were generated by first fitting a model with both surprisal and baseline predictors, and then sampling from that model to show the isolated effect of surprisal alone. I think doing something like this could help make the implications of the data clearer.

(d) (257) I don’t follow this logic. Shouldn’t this mean that the non-linear effects of reanalysis are more apparent in smaller datasets, where a single non-linearity can skew the overall pattern in the data even more?


**Reasons To Accept:**

(a) The authors address a timely topic that has been the focus of recent discussion in the literature

(b) The authors go beyond previous monolingual analyses, which is a valuable contribution

(c) The paper is well written and clear

(d) Although the authors present their handling of spillover has a limitation (I see their point) their method allows for variable spillover between datasets. This is better than many recent papers which set a fixed sized spillover window regardless of dataset.


**Reasons To Reject:**

(a) Previous studies have demonstrated an effect of unigram surprisal (i.e. frequency) on reading times. Frequency is absent from this paper and not used in the baseline model M0. I’m worried that the overall picture of results could change if frequency was added in as an additional predictor. (The authors do mention that word length with random by-participant intercepts was the maximal consistently random effects structure – does this mean that they tried random slopes, or that they tried random slopes plus also other main effects, such as frequency?)

(b) The likelihood ratio tests were computed on the log likelihoods of the training data, correct? I think the paper could be made much stronger if the tests were conducted on a held-out portion of the dataset. Especially as models grow more complex, testing on held-out data grows crucial to avoid charges of overfitting. (My apologies if you are already doing this – it was not clear to me from the text!)


**Reproducibility:**

4: Could mostly reproduce the results, but there may be some variation because of sample variance or minor variations in their interpretation of the protocol or method.

**Reviewer Confidence:**

5: Positive that my evaluation is correct. I read the paper very carefully and I am very familiar with related work.

**Typos Grammar Style And Presentation Improvements:**

(a) (37) “Whereas previous empirical work has addressed this question in English” → I would soften this claim a bit given the recent multilingual work that’s been appearing in the literature. This paper does include new languages, though, so that is worth pointing out up top!

(b) (96) I would mention what a sampling-based model is, briefly, here.

---

> ### Author Rebuttal · Authors · 2023-08-27
>
> 1. We believe that controlling for frequency in these studies is not fully justified: there is good evidence that frequency effects are due only to the correlation between frequency and surprisal (see for example Shain, 2019, NAACL). Nevertheless, in the camera-ready we will include results (in addition to the current results) that include log token frequency as a control.
> 2. Comparing log likelihoods on test data is standard practice in experimental psychology, because a held-out dataset would usually be too small to get a good estimate of the log likelihood. Nevertheless, we will include a heldout analysis (in addition to the current analysis and an analysis using the Wilcox et al 2023 language models).
> 3. For question (a)(135), we will add this information to Table 2. We use first-pass gaze duration whenever possible. All languages except Dutch and German are first-pass gaze duration. Dutch and German are first fixation.
> 4. For question (b)(147), the data exclusions follow previous literature.
> 5. For question (c)(Fig 1), in addition to our current figure, we will include another figure showing residual RT after controlling for word length (and word length spillover). We believe there is value in the current figure, which shows only RT as a function of surprisal, because it is a very simple representation of the data that shows a clearly visually near-linear effect.
> 6. For question (d)(257), we will remove this argument from the camera-ready.
> 7. See our response to Reviewer 2 with respect to Wilcox (2023).
> 8. We will fix all grammar and writing suggestions.

---

### Official Review · Reviewer_JsrJ · 2023-08-04

**Soundness:** 4

**Excitement:**

4: Strong: This paper deepens the understanding of some phenomenon or lowers the barriers to an existing research direction.

**Missing References:**

Andrea de Varda and Marco Marelli. 2022. The Effects of Surprisal across Languages: Results from Native and Non-native Reading. In Findings of the Association for Computational Linguistics: AACL-IJCNLP 2022, pages 138–144, Online only. Association for Computational Linguistics.

**Paper Topic And Main Contributions:**

This paper tests the relationship between surprisal and reading time on seven languages. It is well established that surprisal and reading time are correlated, but recent work has debated whether the linking function is sublinear, linear, or superlinear. Much of the debate has focused on English data; this study expands the investigation to several other languages. Results suggest a slightly superlinear relationship between surprisal and RT.

**Questions For The Authors:**

A) Do you think any of your results would be different if you tested off-the-shelf language models instead of training your own?

B) How did you decide on the stopping criteria in Section 3.2, and why is Mandarin treated differently from the other languages?

**Reasons To Accept:**

The paper contributes useful crosslinguistic data to an ongoing debate about the connection between surprisal and reading time. The methods seem sound and the results are clearly presented.

**Reasons To Reject:**

The opening sections of the paper seem to slightly mischaracterize Surprisal Theory. As presented in Levy 2008, Surprisal Theory posits that comprehension difficulty (e.g. reading time) is proportional to surprisal -- meaning that the linking function is linear. This would mean that theories that posit e.g. a superlinear linking function do not fall under Surprisal Theory per se.

**Reproducibility:**

4: Could mostly reproduce the results, but there may be some variation because of sample variance or minor variations in their interpretation of the protocol or method.

**Reviewer Confidence:**

4: Quite sure. I tried to check the important points carefully. It's unlikely, though conceivable, that I missed something that should affect my ratings.

**Typos Grammar Style And Presentation Improvements:**

A) Wilcox et al. have a very recently submitted preprint which addresses similar questions: https://arxiv.org/abs/2307.03667

B) The Section 2 citation to Hikaru Clark et al. 2023 should just be Clark et al. 2023

---

> ### Author Rebuttal · Authors · 2023-08-27
>
> 1. We will adjust the presentation in the opening section regarding Surprisal Theory. Linear surprisal was posited by Levy (2008) and supported by Smith & Levy (2013), but Hale (2001) does not specify a linking function to RT and Levy's (2005) thesis specifies a monotonically increasing linking function which may be linear or superlinear, which is the presentation we follow here. We will make the citations to the literature more precise to reflect this.
> 2. Initially we experimented with using GPT-3 as the surprisal model for English, finding superlinearity. But we decided against using such models because they introduce uncontrolled and unknowable variation among the languages, in terms of how much and what kind of training data is included for each language. We trained our own models so we could control the perplexity and training data quantity and quality across languages.
> 3. The stopping criteria in Section 3.2 were decided based on compute time available to us. Mandarin is treated separately because, as a byte-level model, its loss is not comparable to the other languages. A token-level model for Mandarin was not feasible due to massive disparities between the LM's tokenization and the RT corpus's tokenization.
> 4. For Wilcox et al. (2023) to appear in TACL and released on the arXiv, in fact, we were not aware of this work, which was posted after the EMNLP submission deadline. Our work differs from that work in several ways and we believe the two papers are complementary:
>     - Wilcox et al. 2023 test on small "sentence corpus"-style datasets of all of their languages, while we use larger corpora of naturalistic text for English, Dutch, and Danish.
>     - Wilcox et al. 2023 do not include analyses of Japanese and Mandarin, which are crucially different from all the other languages in that their orthographies do not have spaces between words. Reading Japanese and Mandarin thus may involve different cognitive processes from the other languages (requiring a segmentation step), so it is interesting that we find linearity for the corpora of these languages.
>     - Wilcox et al. 2023 differ from our work in the statistical methods: their work has a fixed spillover window for all language, whereas ours is variable; their work uses a GAM as the nonlinear model, whereas we use a quadratic regression; and their work controls for frequency whereas our doesn't.
>     - As for the source of the difference in findings (they find linearity for all languages), we believe this will require extensive investigation comparing the two works which goes beyond our current submission. In the camera-ready, we will include a section specifically to compare to Wilcox et al. 2023 to make all the points above.
> 5. Before the camera-ready, we will undertake some additional analysis to make our results more compatible with Wilcox et al. (2023). These will be presented in addition to the current set of results. In particular, we will fit models based on the monolingual LM's from Wilcox et al. (2023) and models controlling for frequency.
> 6. We will fix all grammar and writing suggestions

---

### Official Review · Reviewer_1TwT · 2023-08-05

**Soundness:** 3

**Excitement:**

4: Strong: This paper deepens the understanding of some phenomenon or lowers the barriers to an existing research direction.

**Paper Topic And Main Contributions:**

This paper examines the shape of the function linking next-word prediction with sentence processing load as measured with reading times. The key contribution is an evaluation of this function across seven languages, where previous work has primarily focused on English. This contribution is relevant because it helps to narrow down the precise linking function between a dominant framework for cognitive modeling (surprisal theory) and human behavior.

**Reasons To Accept:**

- The question is of interest for cognitive modeling
- The principle cross-linguistic contribution is important and necessary for this subfield to progress
- Data analysis is well-motivated and connects well with previous work in this domain

**Reasons To Reject:**

- A key limitation concerns the different corpora that were available for the different languages. These corpora differ substantially in size and in genre (e.g. naturalistic text vs constructed sentences). These differences in genre plausibly affect how adequate GPT2 -- given the idiosyncrasies of its own training set -- serves as a cognitive estimator.  The analysis across languages also differ in other ways that may impact the results, including the specific measures available (first-pass gaze duration vs first fixation duration) and the size of the pre-target context that was used to sum "spill-over" surprisal values.

Taken together, I found it difficult to understand where differences between "linear" and "non-linear" effects might reflect cross-linguistic differences, as opposed to idiosyncratic differences of the available corpora (like genre) or other analysis choices.  I think for this sort of work to be maximally impactful, it would be great to see a robustness analysis varying some of these parameters (e.g. corpus size and genre via sub-sampling the available materials; spill-over size etc.)

A final point is that the main conclusions: that languages differ in whether the effect of surprisal is linear or super-linear, is grounded in comparing a set of "significant" model comparison findings against "non-significant" model comparisons. As presented, I don't think this inference goes through ("the difference between significant and non-significant is not itself statistically significant" - German & Stern 2006).  I think this could be addressed by testing for a statistical interaction of language and model-type.

**Reproducibility:**

4: Could mostly reproduce the results, but there may be some variation because of sample variance or minor variations in their interpretation of the protocol or method.

**Reviewer Confidence:**

4: Quite sure. I tried to check the important points carefully. It's unlikely, though conceivable, that I missed something that should affect my ratings.

---

> ### Author Rebuttal · Authors · 2023-08-27
>
> 1. We agree that it is possible that the "by-language" variation is actually by-corpus variation, although we note that we trained separate models on the same genre of text (Wikipedia) for each language. We will rewrite to de-emphasize any claim that language is the main factor influencing the shape of the surprisal effect, and to emphasize the claim of a universal surprisal effect which is linear in many cases and maybe superlinear in others.
> 2. We will remove any conclusion that functional form meaningfully differs as a function of language.

---

### Meta-Review · Area_Chair_66Np · 2023-09-18

**Recommendation:** 4

**Metareview:**

# Overview

This paper investigates the functional form of the surprisal–RT relationship in seven languages. Specifically, it uses likelihood ratio tests to compare linear mixed effect models with: only baseline predictors, baseline + surprisal, baseline + surprisal + surprisal squared. It finds that surprisal helps predict RTs in the 7 studied languages. It also finds that surprisal squared further helps predict RTs in 3 of these languages.

# Meta-review

The three reviewers and I agree this paper tackles an important question, and that its cross-linguistic contribution is important and timely. Further, its methods and results are clearly presented.

The reviewers, however, point out two important limitations in this paper. First, the used crosslinguistic corpora differ substantially in size, genre, and used eyetracking measure; this limits the extent to which conclusions can be drawn. Second, the analysis did not control for frequency effects. (I highlight two more methodological issues below.)

The authors already promised to run an extra analysis with frequency effects for CR (if accepted), and to de-emphasize any claim that language is the main factor influencing the shape of the surprisal effect. I would also suggest they extend the "Eyetracking corpora" paragraph in their limitations section to highlight the issues pointed out by reviewer 1TwT.

# Detailed Comments

While I enjoyed reading this paper, and mostly agree with the three reviewers about its positive aspects, I highlight a few methodological issues below. I believe fixing those could help strengthen this paper.

* **Spillover effects**: While one of the reviewers highlights the way in which this effect was computed as a reason to accept, I think it's actually negative. Specifically, the authors seem to choose spillover effect windows based on likelihood ratio tests using models which only had access to word length. If a spillover effect is significant for surprisal, but not for word length, it will thus not be included in the model. This might have impacted the analysis on Japanese, which at the moment uses a spillover window of 0. There seems to be a large enough number of tokens in most analysed languages for spillover effects to be set to zero by the model fitting procedure itself (if they are insignificant), so I would recommend rerunning experiments (at least for Japanese) with a fixed spillover effect window.

* **Model training**: Models were trained for 7 epochs or until a validation loss of 3.5 nats was reached. Given that not all languages are "equally hard" to language-model (Cotterell et al., 2018; Mielke et al., 2019), stopping the training early (instead of training until convergence) may unfairly harm models in some languages.


* **Frequency effects**: This paper does not control for frequency effects as a baseline feature, using Shain (2019) as a justification. A more recent work by the same author (Shain, 2023) reaches the opposite conclusion. This new paper, however, only came out *after* the EMNLP submission deadline.

The authors did already promise to add an analysis including frequency effects for camera ready, in case the paper is accepted. I would personally also appreciate seeing an analysis using constant spillover effect windows (specially for Japanese), and, if possible, models trained to convergence.

As an aside, in one of their responses, the authors state that Wilcox et al. (2023) tested on small "sentence corpus"-style datasets. I believe Wilcox et al. (2023) used the MECO dataset (similarly to de Varda et al. 2022; 2023), which has paragraph-level entries.

# References

Cotterell et al. (2018). Are All Languages Equally Hard to Language-Model?

Mielke et al. (2019). What Kind of Language Is Hard to Language-Model?

Shain (2019). A large-scale study of the effects of word frequency and predictability in naturalistic reading

Shain (2023). Word Frequency and Predictability Dissociate in Naturalistic Reading

---

### Decision · Program_Chairs · 2023-10-07

**Decision:**

Accept-Findings

**Comment:**

# Overview

This paper investigates the functional form of the surprisal–RT relationship in seven languages. Specifically, it uses likelihood ratio tests to compare linear mixed effect models with: only baseline predictors, baseline + surprisal, baseline + surprisal + surprisal squared. It finds that surprisal helps predict RTs in the 7 studied languages. It also finds that surprisal squared further helps predict RTs in 3 of these languages.

# Meta-review

The three reviewers and I agree this paper tackles an important question, and that its cross-linguistic contribution is important and timely. Further, its methods and results are clearly presented.

The reviewers, however, point out two important limitations in this paper. First, the used crosslinguistic corpora differ substantially in size, genre, and used eyetracking measure; this limits the extent to which conclusions can be drawn. Second, the analysis did not control for frequency effects. (I highlight two more methodological issues below.)

The authors already promised to run an extra analysis with frequency effects for CR (if accepted), and to de-emphasize any claim that language is the main factor influencing the shape of the surprisal effect. I would also suggest they extend the "Eyetracking corpora" paragraph in their limitations section to highlight the issues pointed out by reviewer 1TwT.

# Detailed Comments

While I enjoyed reading this paper, and mostly agree with the three reviewers about its positive aspects, I highlight a few methodological issues below. I believe fixing those could help strengthen this paper.

* **Spillover effects**: While one of the reviewers highlights the way in which this effect was computed as a reason to accept, I think it's actually negative. Specifically, the authors seem to choose spillover effect windows based on likelihood ratio tests using models which only had access to word length. If a spillover effect is significant for surprisal, but not for word length, it will thus not be included in the model. This might have impacted the analysis on Japanese, which at the moment uses a spillover window of 0. There seems to be a large enough number of tokens in most analysed languages for spillover effects to be set to zero by the model fitting procedure itself (if they are insignificant), so I would recommend rerunning experiments (at least for Japanese) with a fixed spillover effect window.

* **Model training**: Models were trained for 7 epochs or until a validation loss of 3.5 nats was reached. Given that not all languages are "equally hard" to language-model (Cotterell et al., 2018; Mielke et al., 2019), stopping the training early (instead of training until convergence) may unfairly harm models in some languages.


* **Frequency effects**: This paper does not control for frequency effects as a baseline feature, using Shain (2019) as a justification. A more recent work by the same author (Shain, 2023) reaches the opposite conclusion. This new paper, however, only came out *after* the EMNLP submission deadline.

The authors did already promise to add an analysis including frequency effects for camera ready, in case the paper is accepted. I would personally also appreciate seeing an analysis using constant spillover effect windows (specially for Japanese), and, if possible, models trained to convergence.

As an aside, in one of their responses, the authors state that Wilcox et al. (2023) tested on small "sentence corpus"-style datasets. I believe Wilcox et al. (2023) used the MECO dataset (similarly to de Varda et al. 2022; 2023), which has paragraph-level entries.

# References

Cotterell et al. (2018). Are All Languages Equally Hard to Language-Model?

Mielke et al. (2019). What Kind of Language Is Hard to Language-Model?

Shain (2019). A large-scale study of the effects of word frequency and predictability in naturalistic reading

Shain (2023). Word Frequency and Predictability Dissociate in Naturalistic Reading